# A Textile EBG-Based Antenna for Future 5G-IoT Millimeter-Wave Applications

**EL May Wissem [1,2], Imen Sfar [1], Lotfi Osman [1] and Jean-Marc Ribero [2,*]**

[1] Department of Physics, Faculty of Sciences of Tunis, University of Tunis El Manar, 2092 El Manar, Tunisia; Wissem.EL-MAY@univ-cotedazur.fr (E.M.W.); imen.sfar.fst@gmail.com (I.S.); lotfi.osman@supcom.tn (L.O.)

[2] LEAT, Université Côte d'Azur, 06903 Sophia Antipolis, France

[*] Correspondence: jean-marc.ribero@univ-cotedazur.fr

**Abstract:** A millimeter-wave (mmWave) textile antenna operating at 26 GHz band for 5G cellular networks is proposed in this paper. The electromagnetic characterization of the textile fabric used as substrate at the operating frequency was measured. The textile antenna was integrated with an electromagnetic bandgap (EBG) structure and placed on a polyester fabric substrate around the antenna. Results showed that the proposed EBG significantly improved the performance of the antenna. The gain and energy efficiency at 26 GHz were 8.65 dBi and 61%, respectively (an increase of 2.52 dB and 7% compared to a conventional antenna), and the specific absorption rate (SAR) was reduced by more than 69.9%. Good impedance matching of the fabricated antenna at the desired frequency was observed when it was bent and worn on the human body. The structure is simple, compact, and easy to manufacture. It may well be suitable for integration into applied clothing in various fields, especially for future IoT applications.

**Keywords:** millimeter-wave; electromagnetic bandgap structure (EBG); 5G; Internet of Things (IoT)

## 1. Introduction

Over the past few years, interest in the millimeter-wave band (mm-wave) has grown rapidly due to the abundance of unused bandwidth and the large available spectrum, open new possibilities for various applications of body-centric wireless communication (e.g., healthcare, entertainment, remote monitoring, mobile computing etc.), that could play a key role in fifth-generation (5G) to connect the billions of fixed and mobile devices for future IoT applications [1].

Wearable antenna features attached to the body or clothing, like size, weight, cost, performance and ease of integration, have been more enormously studied in past years. However, most of these studies have been concentrated on microwave bands [2,3] and more profound efforts are needed for the design and implementation of high-frequency textile antennas to meet the future demand for integrating the IoT with connected clothes. Developing textile antennas at millimeter waves will improve not only data rates but also level of security and reliability. Besides, it should be noted that proposed bands around 20 and 60 GHz bring a challenge for deployment due to propagation and penetration losses with high atmospheric absorption. Hence, to compensate for these losses, high-gain directional antennas are desirable [4].

Recently, the design of textile antennas operating at millimeter-wave has rapidly grown since they guarantee flexibility and comfortable, embedding into clothing while satisfying the requirements of modern wearable systems. Previous studies have been carried out at carrier frequencies to 60 GHz. The first textile Yagi-Uda antenna [5], has been proposed, and shown that the antenna is matched in the 57–64 GHz band for future wireless body area networks (BANs). But, this antenna is suffering from high backward radi-

ation that increases the radiation hazard to human users. Besides, a microstrip patch antenna array printed on textile was proposed for off-body communications in the 60-GHz band and investigated numerically and experimentally [6]. However, the radiation efficiency of this latter was not so promising (40%).

Moreover, a millimeter-wave textile antenna for 5G RF energy harvesting was presented in [7]. The proposed antenna is based on the broadband antipodal Vivaldi antenna covering the 24–28 GHz bands with stable gain over 5 dBi.

However, these antennas for on-body worn applications suffer from certain limitations, contain low radiation efficiency, gain and interaction between antenna and the human body which can significantly affect the performance of the antenna.

Recently, an electromagnetic bandgap material (EBG) realized on a flexible substrate is a standout amongst the best-known method to improve the performance of the antenna [8]. Thanks to its ability to suppress surface waves and generate a zero reflect phase, it offers the potential advantage to improve e gain and quality factor and reduce the effects of e proximity of the human body of the antenna [2,9]. It was shown that the EBG improved the gain, bandwidth and backward radiation suppression of the coplanar waveguide (CPW) antenna in the frequency range 20–40 GHz [10]. Moreover, an EBG based mm-wave flexible multiple-input multiple-output (MIMO) antenna is reported in [11]. The results show that the EBG surface can improve the gain by 1.9 dBi and reduce the backward radiation by 8 dB at 24 GHz. However, these antennas use the EBG structure as a reflector, but these geometries are inherently complex, Thus, their fabrication at millimeter-scale for mm-wave applications is a real challenge. In addition, owing to its multilayer geometry, there are issues of misalignment due to the different parts for assembly, while still retaining a significant lateral dimension. Therefore, a low-profile textile antenna, as well as a single-layer design, have the advantages of high gain and good isolation covering the allocated 5G frequency, which is one of the most requested solutions. In this letter, we provide further results from a study of a mm wave textile patch antenna surrounded by periodic unit cells of EBG printed on a single textile substrate for future 5G and IoT applications [12]. We extend and validate this work through further analysis and by using experimental results. The proposed antenna has an extremely low height profile (0.35 mm) and simple design with minimum parts of the assembly that offer low-cost and mass production suitable for practical applications. Despite the small volume, the proposed antenna generates higher gain and acceptable bandwidth and efficiency with low back radiation and reduced SAR values.

All the antenna simulations were done using computer simulation software (CST Microwave Studio) and fabricated in the Laboratory of Electronics, Antennas and Telecommunications (LEAT) of the University Cote d'Azur, France. The rest of the paper organization is as follows. The dielectric properties of the textile substrate are first characterized at 26 GHz. Then, the patch antenna design, its fabrication process, simulated/measured results and the characterization of the EBG structure are presented. Simulation and experimental results of the EBG-based antennas are compared to conventional designs to appreciate the benefits of EBGs. The on-body performance of the antenna under bending conditions, and SAR assessment, are discussed.

## 2. Characterization of the Textile for the Design of the Antenna

At millimeter waves, the choice and the characterization of the substrate layer in terms of thickness, relative permittivity and loss tangent are very crucial features for any design. In this section, we restrict our choice to the characterization of a well-known textile of 0.35 mm thickness; soft plain-woven polyester fabric.

Different measurement methods were used to determine the dielectric property values, such as the hybrid microstrip-line method, stub resonator and a broadband method based on a stripline cell [13,14]. In this case, the relative permittivity and losses were characterized using the single frequency method based on an open-stub resonator [6,15].

The first step of the proposed methods was to set up calibration by measuring the transmission coefficient $S_{21}$ parameter for a microstrip line printed on the fabric. Next, the $S_{21}$ parameter for quarter-wavelength open stub resonator was measured (see Figure 1a,b), then, the $S_{21}$ parameter for the stub resonator was simulated using CST Microwave Studio . The relative permittivity and the dielectric losses of the textile were adjusted to match the simulated $S_{21}$ curve with measured one (Figure 1c).

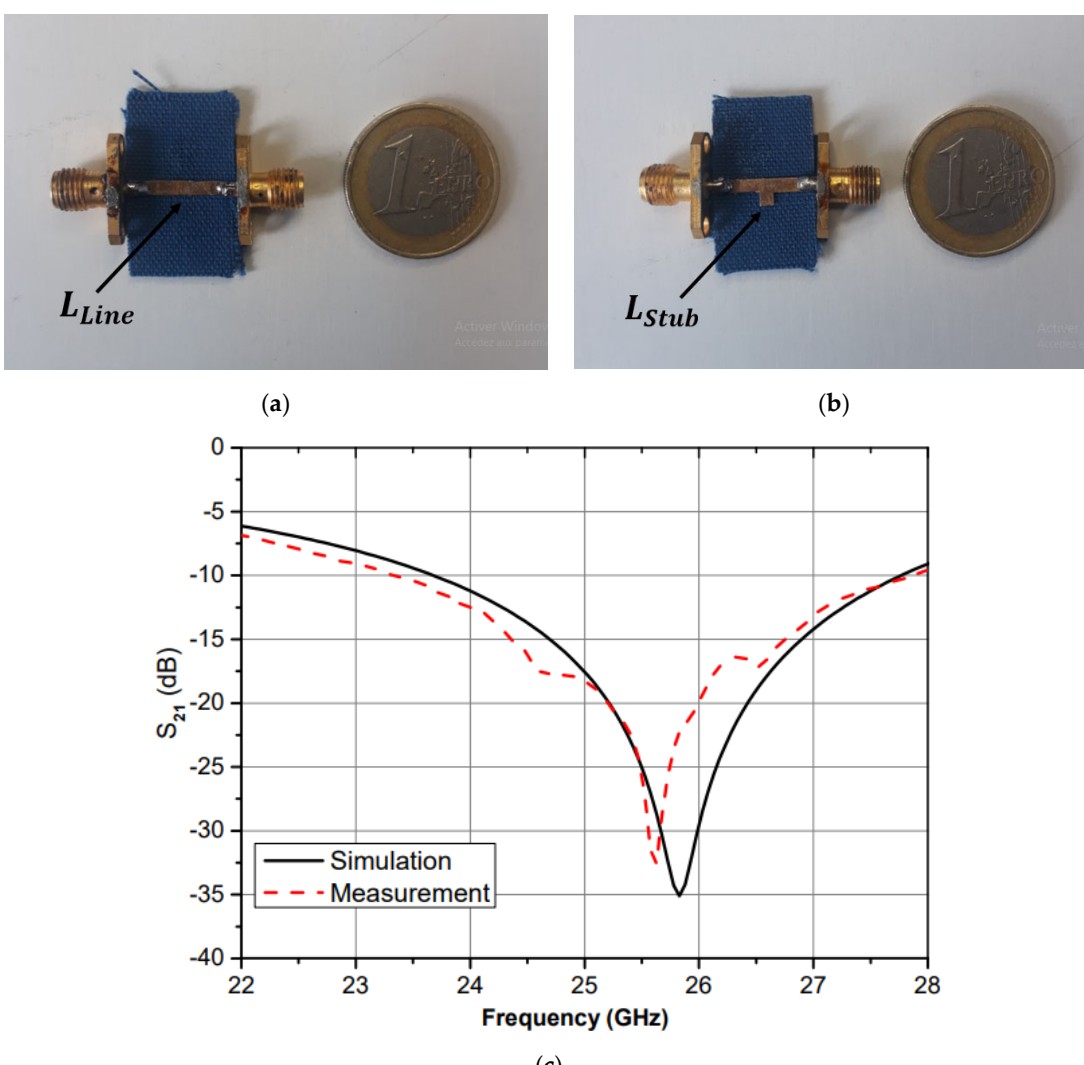

**Figure 1.** Characterization of the polyester fabric at 26 GH. (**a**) Prototypes of the microstrip line and microstrip T-resonator. (**b**) Set-up of S-parameter measurement. (**c**) Simulation and measurement results of $S_{21}$ parameter.

The proposed technique offers many advantages in terms of rapidity, simplicity and no requirement for specific equipment in addition to its perfect adaptation to all types of fabrics. The 50 Ω microstrip line width and open-stub length were equal to 1.10 mm and 1.90 mm, respectively. It was bonded on a piece of polyester fabric (10 × 18 × 0.35 mm). The metallization of the stub and the ground plane was realized by adhesive copper foil.

The best agreement for the resonant frequency was achieved using $\varepsilon_r$ = 2.17 and tan δ = 0.0035. These values were used in the design of the antenna.

## 3. Antenna and EBG Design

### 3.1. Antenna Design and Fabrication

Figure 2a shows a conventional patch antenna which consists of a conductive layer and 0.35 mm-thickness of the polyester textile substrate layer. The rectangular patch antenna is the easiest to construct and was fed by a microstrip transmission line to facilitate the integration into the textile. The dimensions of the antenna were calculated using equations cited in [16] and are listed in Table 1.

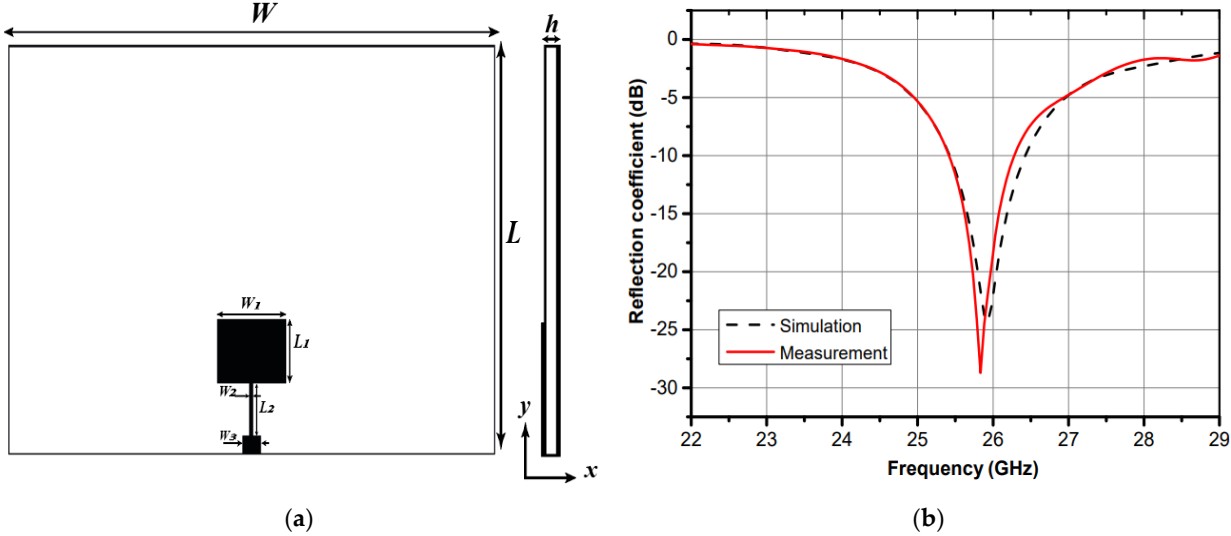

(**a**)                    (**b**)

**Figure 2.** Conventional textile antenna. (**a**) Dimension of the textile antenna. (**b**) Simulation and measurement of the return loss $S_{11}$.

**Table 1.** Dimensions of the proposed antenna.

| Parameter | Value (mm) | Parameter | Value (mm) |
|-----------|-----------|-----------|-----------|
| W | 23 | $W_2$ | 0.2 |
| L | 18 | $L_2$ | 2.04 |
| $W_1$ | 4.46 | $W_3$ | 1 |
| $L_1$ | 3.3 | h | 0.35 |

The radiating element and feeding line were fabricated by using ultra-thin polyimide copper laminates [17,18]. The etched polyimide copper laminates present good flexibility compared to copper foils and the manufacturing imperfections were minimized, as mm wave antenna dimensions are generally less than 1 mm. The nonpatterned metallic parts (like the ground plane) were made of a thin layer of copper foil with adhesive backing. Figure 3 shows a cross-section of the structure.

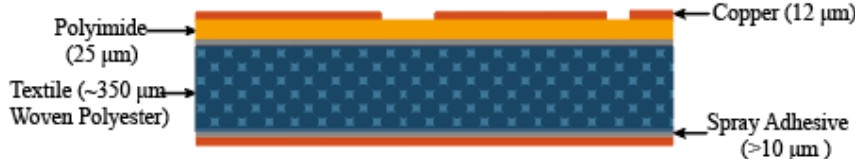

**Figure 3.** Cross-section of assembled structure on textile.

The reflection coefficients frequency comparison between the measured and simulated results of the conventional textile antenna are shown in Figure 2b. Both results were

in good agreement at 26 GHz, whereas the measured bandwidth was 0.91 GHz, which is 13.3% less than the predicted bandwidth of 1.05 GHz. This is because of the imperfections reproduced during the fabrication process (soldering tolerance). The radiation patterns of the antenna are described later, and the measured gain of this latter was 6.13 dB at 26 GHz.

Whereas an on-body antenna in practice needs to be not only compact and lightweight for better integration, it should also be efficient and induce minimal power absorption inside the human body. Therefore, the surface waves excited by the edges of the antenna can considerably deteriorate the performance of the antenna, such as radiation efficiency and gain that may not satisfy anticipated future requirements.

To overcome this problem, a band-gap material (EBG) structure was integrated into the design. Such metamaterial structures have two common configurations: in-phase reflection and surface waves suppression due to the property of high impedance surface (HIS) within a certain frequency band gap in which surface wave propagation is highly restricted [19]. Figure 4 presents a generic flowchart, which explains the process of the textile antenna design starting with the characterization of the fabric using as a substrate. The EBG structure is patterned around a conventional patch antenna for performance enhancement. Thereafter, measurement of the fabricated antenna was conducted. In addition, further evaluation of the textile antenna on the human body was tested.

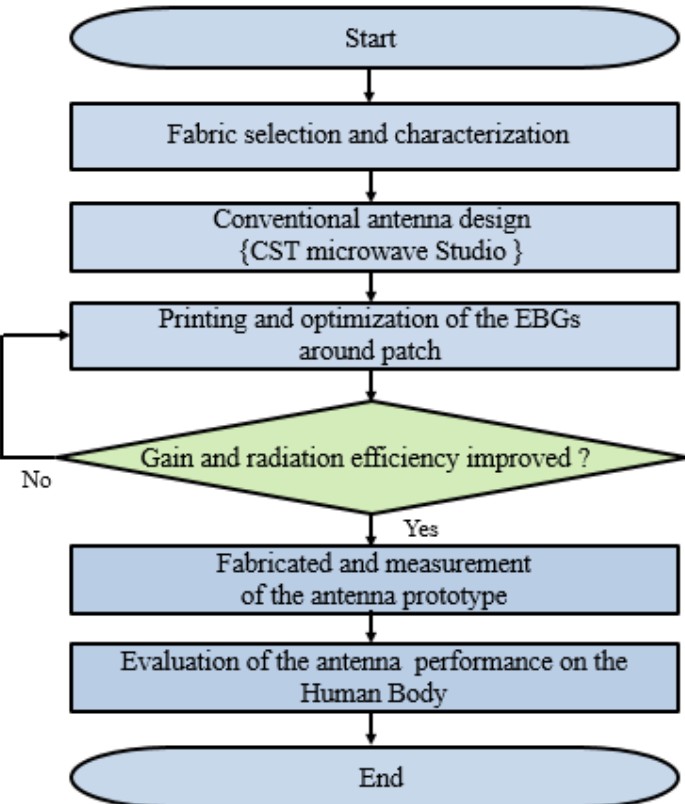

**Figure 4.** The flowchart describing the design procedure of the proposed textile antenna.

### 3.2. EBG Design

The EBG unit cells have been reported in [20]. They were used to surround the patch radiator, which offers the advantage of reducing the surface wave influence. Surface-waves suppression can reduce the quantity of power wasted, and back radiation leads to improved antenna performance aspects such as raising the gain, side-lobe and back-lobe reduction, and improving the energy efficiency [21].

The size of the proposed EBG unit cell is shown in Figure 5. The fabric used was polyester and the square patch was fabricated with the same copper sheets.

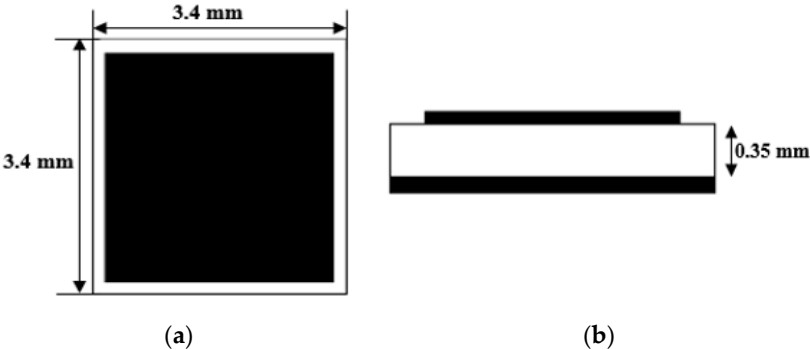

(**a**) (**b**)

**Figure 5.** Electromagnetic bandgap (EBG) unit cell geometry. (**a**) Front view; (**b**) bottom view.

Diverse methods were employed to investigate the EBG characteristics at the resonant frequency, such as the dispersion diagram, reflection phase [22] and suspended transmission line method. In this project, the suspended transmission line method was considered. This technique has been usually used and gives many advantages such as weak transmission loss, low dispersion characteristics and ease of fabrication [23]. For the proposed technique, an insulating microstrip transmission line was hung over the EBG array and excited such that one port acted as a source and the other as a matched load. The bandgap response was calculated with the optimized dimension of the square unit cell EBG using the simulator CST MWS as shown in Figure 6.

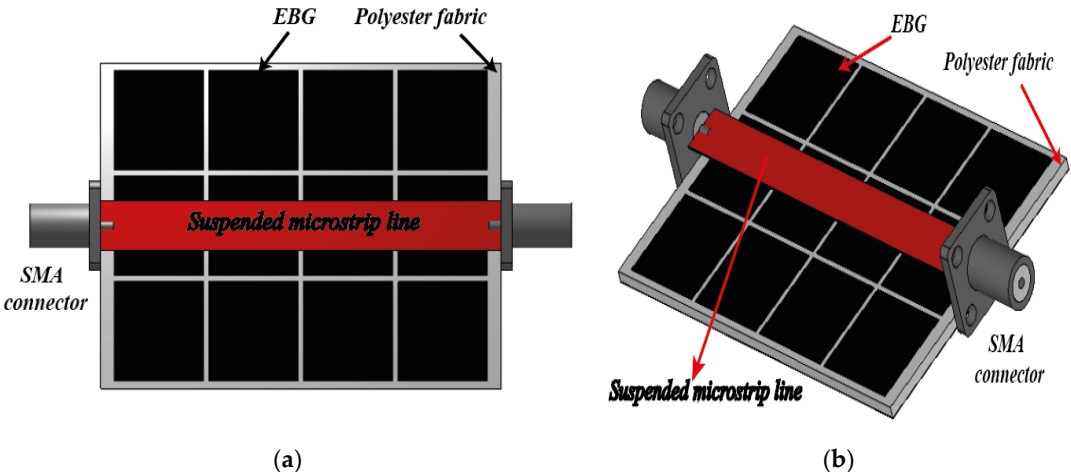

(**a**) (**b**)

**Figure 6.** Suspended microstrip line method of EBG: (**a**) top view; (**b**) 3-D view.

The simulation results are depicted in Figure 7. The transmission coefficient ($S_{21}$) presented a sufficiently well-defined bandgap in the frequency range from 25.53 GHz to 28.12 GHz. After characterization, the EBG unit cell was used to surround the patch antenna for the surface-wave suppression and helped to improve the performance of the antenna [20].

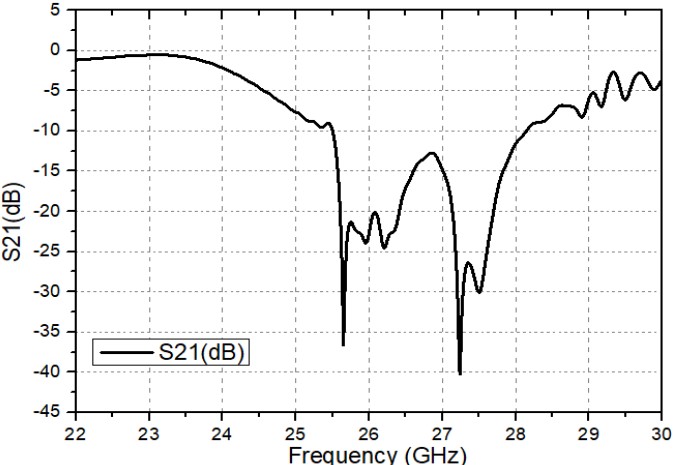

**Figure 7.** Simulated ($S_{21}$) bandgap response EBG using a suspended line method.

### 3.3. Integration of Conventional Antenna and EBG Surface

In this section, the EBG unit cells were loaded in the top layer of the three-sided rectangular patch antenna as shown in Figure 8a. The top face was made up of fourteen EBG unit cells which were placed in an optimal position away from the edge of the antenna. The other two sides (right and left) consisted of nine periodic squares of EBG unit cells and were placed within one period of the edges of the patch. This arrangement between the antenna and the EBG layers was made to obtain a better antenna return loss.

The new antenna had dimensions of $32.1 \times 22$ mm$^2$. The size of the EBG unit cell was $3.2 \times 3.2$ mm$^2$ and the space between the two cells was 0.2 mm.

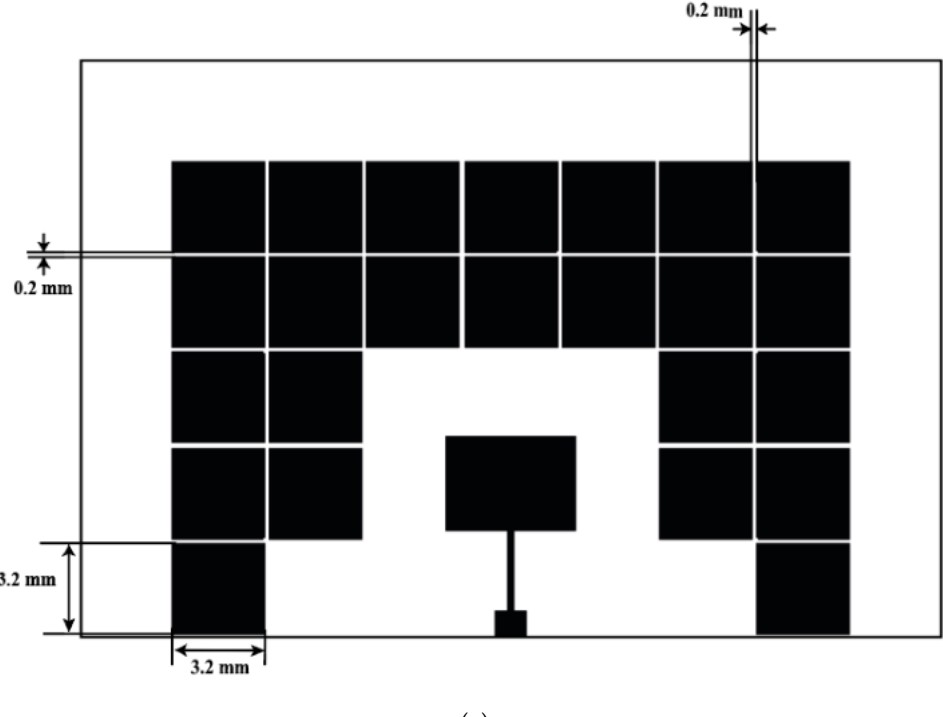

(**a**)

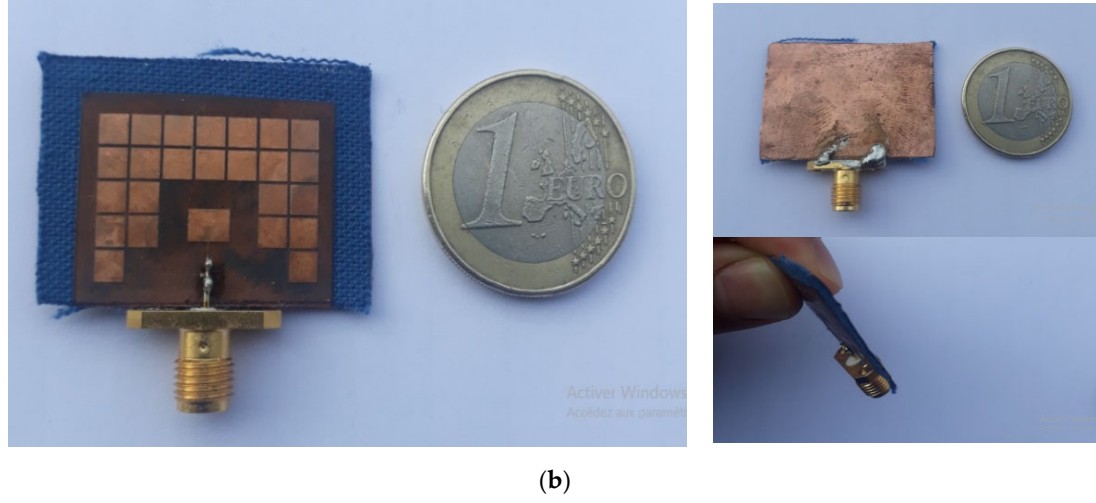

(**b**)

**Figure 8.** Geometry of the proposed antenna: (**a**) top view of the patch with EBG; (**b**) fabricated antenna.

## 4. Discussion

The fabricated antenna prototype is shown in Figure 8b. To explain the antenna performance, the reflection coefficient plot of the antenna measured in free space with and without EBG is depicted in Figure 9.

We can notice that the antenna with EBG offers a good impedance matching ($|S_{11}|$ < −10 dB) for a bandwidth frequency ranging from 25.51 GHz to 26.27 GHz, which corresponds to a bandwidth of 760 MHz. This value is sufficient for 5G−26 GHz operations.

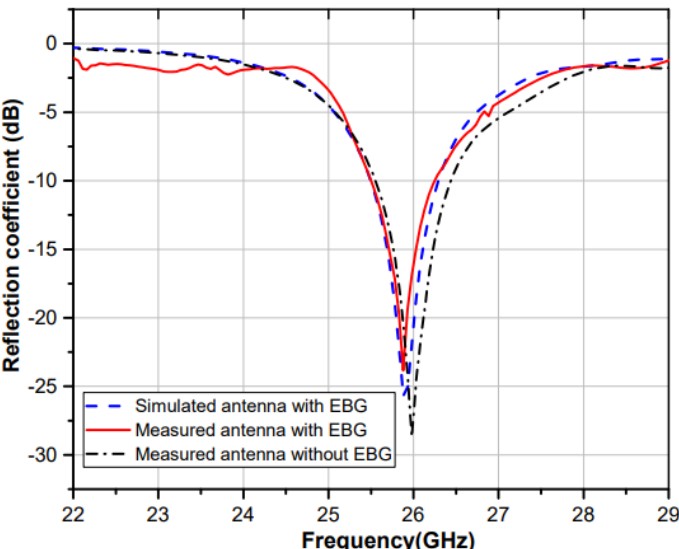

**Figure 9.** Return loss plot of the antenna measured in free space.

The measured radiation pattern plots of the two designed textile antennas in both E and H planes were obtained in an anechoic chamber as shown in Figure 10a. From the results of Figure 10b, it can be seen that the textile antenna loaded with EBG significantly narrowed the main radiation beam compared to the antenna without EBG. In particular, the half-power beamwidth (HPBW) in the E-plane was substantially reduced by 33.2°, thus enhancing its radiation directivity and gain.

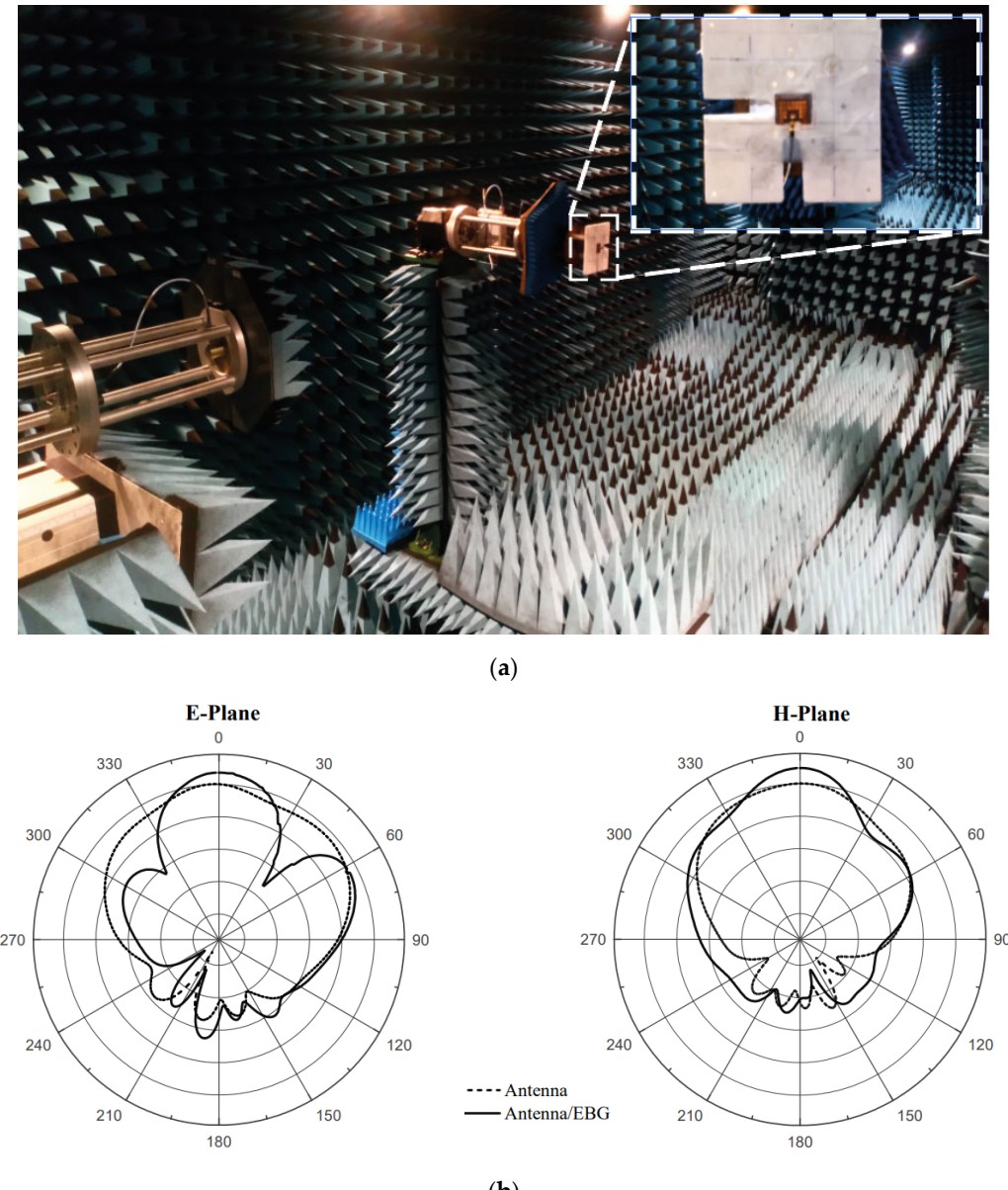

**(a)**

E-Plane

H-Plane

---- Antenna

—— Antenna/EBG

**(b)**

**Figure 10.** Radiation patterns: (**a**) in the anechoic chamber. (**b**) measured for two planes (E-plane; H-plane) radiation patterns at 26 GHz.

Table 2 shows the directivity, radiation efficiency and gain of the proposed and conventional patch antennas, both at their operating frequency. It can be seen from this table that the directivity and the measured gain of the antenna with EBG increased by 2 dBi compared to the classic patch antenna without EBG. As a result, the measurement results of the antenna achieved an improved radiation efficiency of about 7% when EBG was added.

**Table 2.** Radiation parameters of the two patch antennas at 26 GHz.

| Antennas | without EBG | | with EBG | |
|---|---|---|---|---|
| | Simulated | Measured | Simulated | Measured |
| Gain (dBi) | 6.97 | 6.13 | 9.79 | 8.65 |
| Directivity (dBi) | 7.71 | 8.3 | 11.1 | 10.4 |
| Efficiency (%) | 72 | 54 | 77 | 61 |

## 5. Antenna on the Human Body

### 5.1. Bending the Antenna

In mobile systems, a wearable antenna made of flexible textile is expected to bend due to the movements of the human body. The proposed textile antenna with EBGs was measured when bent over a rolled foam of 60 mm diameter along the x-axis and y-axis, thus modeling the forearm of an adult human, to study the performance of this latter under deformation, as can be seen in Figure 11a. The measured $S_{11}$ when bent in both planes in the x and y axis are plotted in Figure 11b and compared to the when the antenna was flat. It is clear from the results that the resonant frequency of the antenna when bent increased slightly and the bandwidth remained the same. It can be concluded that the EBG textile antenna remained in an acceptable frequency range even when bent.

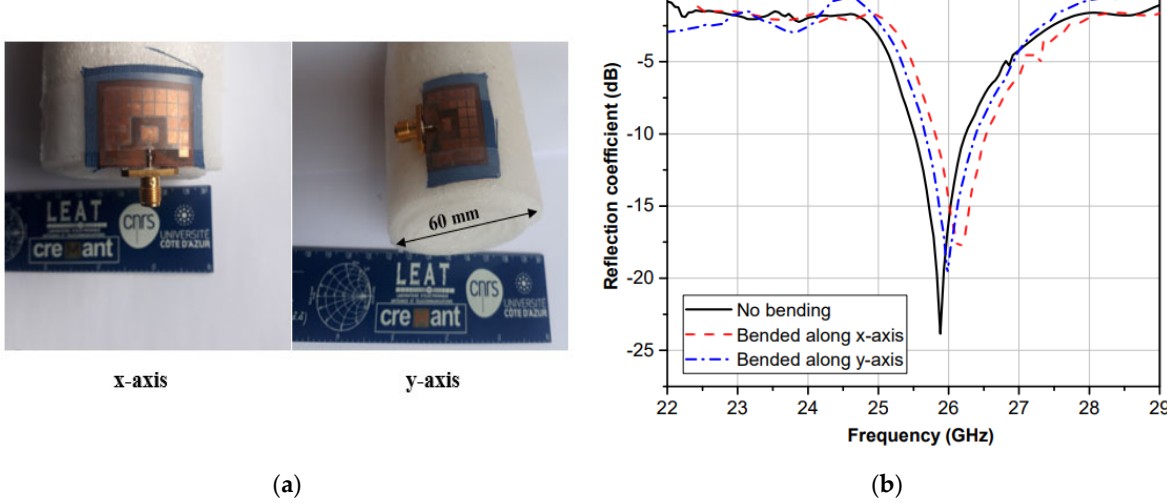

(**a**)                      (**b**)

**Figure 11.** Performance of the proposed antenna when bent; (**a**) bending of the EBG textile antenna; (**b**) measured return loss $S_{11}$.

### 5.2. Effects of Human Tissues on Reflection Coefficient

To verify the antenna performance on the human body, the fabricated antenna with EBG was placed and tested on different parts such as the forearm, leg and arm of a real male volunteer who weighed 90 kg and was 183 cm in height as shown in Figure 12a. The measured $S_{11}$ curves of the proposed antenna on the body are plotted in Figure 12b. There was a slight difference between the curves due to the high dielectric nature of human tissue. However, the curves and the bandwidths of $|S_{11}| < -10$ dB covered the desired band 26 GHz in all conditions.

### 5.3. SAR Evaluation

To further investigate the interaction between the antenna and the human body when exposed to an electromagnetic field, the specific absorption rate (SAR) of the antenna without and with EBG were simulated and compared in CST MWS using a simplified

human model. To calculate the specific absorption rate value, the following formula was used:

$$SAR = \sigma \frac{|E|^2}{\rho}$$

where $\rho$ is the volume density of the human, $\sigma$ is the conductivity of human tissues and E is the electric field.

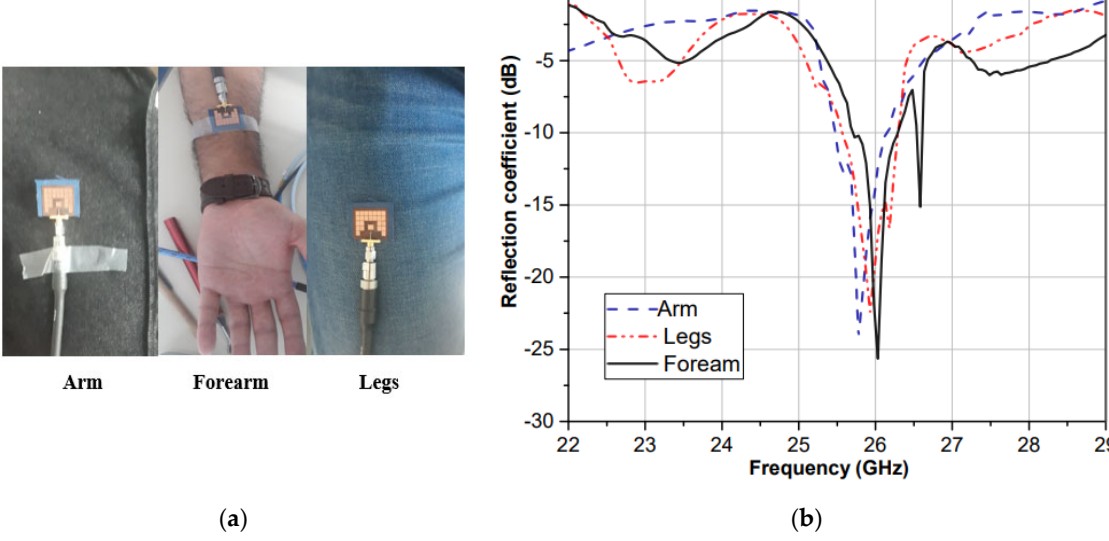

(a) (b)

**Figure 12.** Antenna measured on the body: (**a**) performance evaluation of the antenna at the arm, forearm, and legs; (**b**) measured return loss $S_{11}$.

The human tissue model was added below the antenna and was made using a rectangle with three layers of skin, fat and muscle. The dielectric properties of the layers, which vary with frequency, were retrieved from the works of the Italian National Research Council, which is available online [24] and are tabulated in Table 3. The size of the model was $50 \times 50 \times 47$ mm$^3$, and the distance between the model and the prototype was 1 mm.

The SAR value of the antenna with and without EBG using the IEEE C95.3 averaging method for 10 g of tissue volume were 0.096 W/Kg and 0.32 W/Kg, respectively. In both cases, the maximum SAR value was less than the safe level for the European standard 2 W/Kg thanks to the presence of the large area of the ground plane which reduced the backward radiation, whereas in the EBG case, the results were better; 69.9% lower than those without EBG.

**Table 3.** Material properties of the human body model at 26 GHz.

|  | Dielectric Constant $\varepsilon_r$ | Loss Tangent tanδ | Conductivity σ (S/m) | Thickness (mm) |
|---|---|---|---|---|
| Dry skin | 19.78 | 0.86 | 24.74 | 2 |
| Fat | 3.76 | 0.29 | 1.59 | 5 |
| Muscle | 25.84 | 0.84 | 31.59 | 40 |

## 6. Conclusions

A mm-wave textile antenna intended to be integrated into clothing for future 5G and IoT applications was presented in this article. The dielectric properties of the textile fabric used as a substrate were characterized at 26 GHz using the open-stub technique. A description of the design and manufacturing procedure was detailed. The proposed textile antenna was surrounded by a layer of electromagnetic bandgap structure. The introduction of this type of structure improved the gain by 2.52 dBi and the radiation efficiency by

7%. Finally, bending and an on-body scenario measurement carried out with the prototype produced, led to good results in terms of radiation parameters and specific absorption rate.

The prototype can be seen as a very promising solution for 5G and IoT applications of the mm-wave range thanks to its simple geometry, high gain and good ability to function when folded and worn on the human body.

**Author Contributions:** E.M.W. made significant contributions to this study regarding conception, design, and analysis and writing the manuscript. J.-M.R. and L.O. supervised the whole study and participated in revising the article critically for important intellectual contents. I.S. also supervised the study. All authors have read and agreed to the published version of the manuscript.

**Funding:** This research received no external funding.

**Institutional Review Board Statement:** Not applicable.

**Informed Consent Statement:** Not applicable.

**Data Availability Statement:** Data sharing not applicable. No new data were created or analyzed in this study. Data sharing is not applicable to this article.

**Conflicts of Interest:** The authors declare no conflict of interest.

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
