# Peer review of "A Textile EBG-Based Antenna for Future 5G-IoT Millimeter-Wave Applications"

_electronics, doi:10.3390/electronics10020154_

Round 1

Reviewer 1 Report

* Plots in general: please consider using vector graphics (PS/SVG/PDF) for the plots.

* Photographs: Please use higher resolution/less strongly compressed pictures. Important details are barely (or not) discernible with the supplied photographic material.

* Fig. 1: Please place the text for both pictures outside the device-under-test (L_line).

* Lines 83-87: The authors state that the curves are "digitally adjusted" to match. How was this "digital adjustment" performed?

* Lines 98/99: How was the bonding performed?

* Line 114: how thick is the Polyimide substrate? It would be very useful to include a sketch of the complete antenna assembly.

* Fig. 6: Please include a photograph of the bottom and the side of the antenna. Where is the ground plane located? This is a very important detail that is unfortunately not discernible on any photo.

* Line 180: The word "impedance" is probably misplaced here.

* Fig. 8: Please correct the figure titles: "E-Plane" and "H-Plane".

Reviewer 2 Report

Comment 1

Part of the article is a duplication of another article by the authors.

Comment 2

Please indicate what the presented solution is better or new compared to the already known EGB solutions.

Comment 3

The foam used is well suited to simulating the curvature of the human body, but is it a good model in terms of electrical parameters? (line 207,208). The foam was only used to study the effect of curvature? Without taking into account the electrical parameters of the human body?

Reviewer 3 Report

  1. The introduction section or related works should discuss the past research gaps. Why you should study this issue? What are your research goals? Do you have solved these problems?
  2. The authors do not make a comparison with related works. It is hard to persuade readers.
  3. Your method should use a flowchart to illustrate the processing process.
  4. The references do not conform the journal format.

Round 2

Reviewer 3 Report

1.The authors do not respond the point 2 of the previous concerns. Moreover, the authors do not use the elliptic graph to present the "start" and "end" in Fig. 4.

2. Overall, the quality of the presentation is not good. 

Author Response

in the box if you only upload an attachment
